# Subjective and objective difficulty of emotional facial expression perception from dynamic stimuli

Jan N. Schneider[1,2]*, Magdalena Matyjek[2¤a]*, Anne Weigand[2¤b], Isabel Dziobek[2], Timothy R. Brick[3]

**1** Institut für Informatik und Computational Science, Universität Potsdam, Potsdam, Germany, **2** Department of Psychology, Berlin School of Mind and Brain, Humboldt-Universität zu Berlin, Berlin, Germany, **3** Department of Human Development and Family Studies and Institute for Computational and Data Sciences, The Pennsylvania State University, State College, PA, United States of America

¤a Current address: Center for Brain and Cognition, Universitat Pompeu Frabra, Barcelona, Spain
¤b Current address: MSB Medical School Berlin, Berlin, Germany
* jan_schneider@live.de (JNS); magdalena.matyjek@hu-berlin.de (MM)

**Data Availability Statement:** Data are available in an anonymised form at https://osf.io/vuktj/.

**Funding:** This research was funded by the German Federal Ministry of Education and Research

## Abstract

This study aimed to discover predictors of subjective and objective difficulty in emotion perception from dynamic facial expressions. We used a multidimensional emotion perception framework, in which observers rated the perceived emotion along a number of dimensions instead of choosing from traditionally-used discrete categories of emotions. Data were collected online from 441 participants who rated facial expression stimuli in a novel paradigm designed to separately measure subjective (self-reported) and objective (deviation from the population consensus) difficulty. We targeted person-specific (sex and age of observers and actors) and stimulus-specific (valence and arousal values) predictors of those difficulty scores. Our findings suggest that increasing age of actors makes emotion perception more difficult for observers, and that perception difficulty is underestimated by men in comparison to women, and by younger and older adults in comparison to middle-aged adults. The results also yielded an increase in the objective difficulty measure for female observers and female actors. Stimulus-specific factors–valence and arousal–exhibited quadratic relationships with subjective and objective difficulties: Very positive and very negative stimuli were linked to reduced subjective and objective difficulty, whereas stimuli of very low and high arousal were linked to decreased subjective but increased objective difficulty. Exploratory analyses revealed low relevance of person-specific variables for the prediction of difficulty but highlighted the importance of valence in emotion perception, in line with functional accounts of emotions. Our findings highlight the need to complement traditional emotion recognition paradigms with novel designs, like the one presented here, to grasp the "big picture" of human emotion perception.

("Bundesministerium für Bildung und Forschung") as part of the EMOTISK project (funding number 16SV7240K). The funders had no role in study design, data collection and analysis, decision to publish, or preparation of the manuscript.

## Introduction

Facial expressions are an essential part of human interaction and have been a research topic for more than a century [1]. Facial expressions transmit information between the individual displaying the expression (called the actor in the following) and the individual perceiving the expression (called the observer in the following). For facial expressions to function as an effective communication channel, the actor must produce expressions that will be readily understood by the observer and the observer must possess the necessary abilities to understand the expressions appropriately. Characteristics that have an influence on this transmission process, for example the age or sex of the actor and observer, typically have been studied with emotion recognition paradigms [e.g., 2,3]. *Recognition* means matching a stimulus to some concept recalled from memory, which, in the case of emotions, implies a distinct set of predefined emotion categories and a formally correct answer to recognize. However, the validity of this approach has been challenged by recent research.

In this paper, we first discuss the limitations of common emotion *recognition* paradigms: the use of discrete basic emotion categories, low ecological validity of stimuli, and *a priori* "ground truth". Instead, we present an argument for the validity of studying emotion *perception*, i.e., all perceptual processes that occur when an observer views an expression. We identify potential factors contributing to difficulties in this perceptual process and propose a novel paradigm in which ground truth is defined by population consensus in a dimensional space of emotion. Using this novel paradigm, we investigate predictors of subjective and objective difficulty of emotion perception in a sample of 441 participants rating dynamic emotion stimuli (videos with actors displaying different facial expressions).

### Limitations of common emotion recognition paradigms

**Discrete response labels.**   The force-choice paradigm has been commonly used in studies investigating recognition of facial expressions. In this paradigm, only a single discrete emotion category is predefined as the correct answer [e.g., 4,5], most often the category the actor intended and/or was instructed to display. The discrete categories of the stimuli as well as the response options were commonly selected to align with one of Ekman's basic emotions [6]: a set of six prototypically displayed emotions; This choice for stimuli implicitly assumes that emotions are perceived to be discrete and exclusive. However, studies [7,8] have repeatedly shown that even affective states of opposite valence (positive and negative affect, or disgust and amusement) can be experienced (and expressed) at the same time. Thus, an alternative response option to discrete categories is framing the emotion space as multidimensional: This approach assumes that the complex phenomenon of emotion can be represented by a number of meaningful dimensions, in much the same way that the color space can be represented with brightness, hue, and saturation (the number of such dimensions is a matter of scientific discussion; [9]). Indeed, studies showed that participants readily use multiple emotions in a dimensional way to describe even prototypical target emotions [10–12]. This evidence supports a multifaceted and dimensional, rather than an exclusive and categorical, perception of emotion. It could also be seen as supportive of the concept of blended emotions, i.e., more complex emotions composed of basic emotions as their component parts [13].

Moreover, it is possible that the reduction of emotional perception into discrete labels occludes true effects and produces artifacts. In a study by Hall and Matsumoto ([11], Study 2), increased accuracy of female participants was only found when dimensional response options instead of discrete emotion labels were provided. This indicates that some group differences may lie in the pattern of multi-dimensional emotion judgments and thus cannot be observed when participants' response is limited to a single choice from a few predefined emotion labels.

Further, in a study on the commonly used forced-choice basic emotion paradigm, Frank and Stennett [14] found wide agreement on an incorrect label when the correct one was omitted, and even for a nonsensical expression for which a response label was not conceivable. They showed that they could remedy this effect by providing a "none of these terms is correct" option. This research questions the generalizability of effects found in forced-choice paradigms.

**Ecological validity of stimuli in emotion recognition paradigms.** Common basic emotion stimuli sets often suffer low ecological validity. For example, they frequently include prototypical and over-exaggerated facial expressions that produce ceiling effects and that lack ecological validity [5]. In fact, some effects, such as improved emotional facial expression perception among females were only detected when stimuli were used in subtle and "toned-down" versions [3,4]. Past studies traditionally employed static stimuli in the form of photographs. However, in recent years, research has consistently reported that emotional facial expressions have a specific dynamic time signature which is perceived as most naturalistic [15,16] and which results in higher recognition accuracy compared to deviant time signatures [17]. Further, dynamic facial expressions are recognized faster and with higher accuracy than static ones [18]. Differences are also found on the neural level: Distinct neural pathways [19] and brain responses measured with EEG [20] were found for the processing of static versus dynamic stimuli, even when dynamics were perturbed but not removed. A review by Krumhuber, Kappas and Manstead [21] concluded that dynamic information increases emotion recognition accuracy, ratings of intensity and arousal, and detection rates of genuine and fake expressions; a finding backed by automated expression analysis [22]. They argued that dynamics are an important part in understanding the phenomenon of facial expressions and urged researchers to overcome the use of static facial expression stimuli. Together, these suggest that to be ecologically valid, research should include stimuli that are comparable to authentic real-life expressions, specifically those showing naturalistic and dynamic representations of facial expressions.

**The question of ground truth.** A "ground truth" for a given emotion can be defined as "an emotion that is perceived by people and that is agreed upon by most of the receivers" [23]. Traditional studies tend to assume that the intention of the actor determines the meaning of their emotional expression. Consequently, posed expression stimuli are often labeled with the emotion that was requested from the actor [24]. In this case, accuracy in a emotion recognition paradigms refers to the proportion of labels assigned by the participant that matches these predetermined stimulus labels (i.e., an *a priori* ground truth). However, a facial expression might not always convey its intended meaning. For example, a person smiling at a camera might appear awkward or uncomfortable rather than showing the intended emotion of happiness. Thus, a potentially more appropriate ground truth for an emotional label is the consensus among some set of observers that share cultural similarities. This raises a new question in the context of group differences. In the traditional approach, if two groups differ systematically in their interpretations of an expression, at least one group is considered to be incorrect. If the ground truth reflects a cultural consensus, however, it is difficult to argue that the consensus among, for example, male observers is more or less correct than the consensus among female observers. Instead, these should be interpreted as reflective of different biases and therefore different ground truths in these two groups. In this new light, traditional studies seem to employ a notion of ground truth which is too rigid. This motivates the need for a novel research paradigm, which would allow researchers to utilize a group's consensus as the ground truth and deviations from this consensus as difficulties in perception. Here we propose such a paradigm and focus on identifying predictors of the difficulty of emotional facial expression perception.

## Difficulty of emotion perception

**Subjective and objective difficulty.** It is evident that some facial expressions are generally easier to understand than others. For example, within Ekman's basic emotions [6], expressions of happiness show a higher recognition accuracy across cultures than expressions of disgust or fear [25]. However, it is unclear what features of the actor, emotion, or observer lead to difficulty in perceiving an emotional facial expression. Difficulty might also take on several forms. A traditional objective measure of difficulty focuses on how often participants are *incorrect* in interpreting the expression; in continuous emotion perception, this would correspond to the size of the deviation from the cultural consensus. On the other hand, an observer might also *report* that an expression was more difficult to interpret (regardless of how close their rating was to the cultural consensus), providing a second, potentially independent, measure of difficulty. The two facets of difficulty can therefore be examined separately: *subjective* difficulty as reported by the observer and *objective* difficulty reflecting the observer's abilities in relation to the rest of the population.

Although to our knowledge subjective and objective difficulty in emotion perception have not been yet addressed systematically in research, some insights into their interplay are provided by studies correlating participants' confidence about their responses and the actual performance. For example, in a study by Kelly and Metcalfe [26], participants' trial-by-trial confidence in providing a correct answer in an emotion recognition task has been found to correlate with their performance, but their self-assessments of general emotion recognition abilities (interpersonal sensitivity, empathy, ability to read and interpret mental states and intentions of others) did not predict task performance. Significant rank-order correlations between trial-by-trial performance and confidence self-reports in this study were in the range between .07 and .45, which suggests a weak to moderate effect. Higher correlations were found in the Ekman Emotional Expression Multimorph Task [27], which features whole-face stimuli, than in the Reading the Mind in the Eyes task [28], which features only the eye region; Correlations were also higher when response options were revealed next to the stimuli. Clearly, the amount of information given about the emotion task is a crucial factor in accurately predicting whether a "correct" answer will be provided. Overall, these findings indicate that the subjective and objective difficulties in emotion perception are not strongly correlated. We therefore theorize that different measures of difficulty capture influences from different aspects of the human emotion perception process and thus aimed to evaluate them separately in this study.

**Predictors of subjective and objective difficulty.** In this study, we quantified the difficulty of emotion perception in two different ways. The subjective difficulty of evaluating a given emotional stimulus was measured by observers' self-reported difficulty rating. Objective difficulty was operationalized as the deviation of an individual's perception of a given emotion from the cultural consensus among perceivers. For this, we estimated a *consensus point* in a dimensional emotion space with the mean ratings given by a sample of observers. Importantly, this measure was estimated from the study sample (see Methods for details on calculation) and was therefore free from biases stemming from researchers' decisions on stimulus labels. Here, we briefly review potential person- and stimulus-specific factors which the literature suggests may contribute to both subjective and objective difficulty in emotion perception.

*Person-specific.* Age and sex of both actors and observers have been implicated as modulators of emotion perception. For example, a review by Fölster, Hess, and Werheid [29] concluded that emotional expressions are more difficult to read from old faces than from young ones. These differences might stem from decreased intentional muscle control in the elderly, affecting posed emotions and negative implicit attitudes towards old faces. It has also been shown that morphological changes of the face, such as folds and wrinkles, interfere with the

emotional display in the elderly [30]. Further, women are more expressive [31,32], for example showing more smiles and more frequent head movements than men [33–35]. In a similar vein, a recent study used automatic facial expression recognition technology on over 2000 participants from five countries who displayed emotional expressions while watching television advertisements and showed significant sex differences in emotional expressivity [36]. In particular, women displayed most of the investigated facial actions more frequently than men, although men showed a greater frequency of brow furrowing compared to women.

In observers, emotion recognition capabilities have been reported to improve consistently with age until late adolescence [37–39]. However, there is also an agreement that decoding abilities decline with increasing age in older adulthood [40,41]. Sex of the observer also plays a role in the perception of emotional expression. A meta-analysis concluded that women have a small advantage over men in the recognition of non-verbal displays of emotion (mean Cohen's d = 0.19; [42]). Other previous studies have found sex differences in recognition abilities only when employing subtle emotional stimuli [3,4], which again emphasizes the need for ecologically valid stimuli in experimental research.

*Stimulus-**specific**.* The well-known core affect model [43,44] describes the space of emotions in terms of *valence*, which captures the level of positivity or pleasure of a stimulus, and *arousal*, which describes the level of alertness or physical activation. These dimensions are widely used as the underlying variables in dimension reduction analyses (such as factor analysis or multidimensional scaling) on self-reported or ascribed affect and ordered emotional words. Moreover, they have been indicated as core factors in dimensional descriptions of emotions [9]. Hence, in this study we investigated the relationship of emotion stimuli's valence and arousal with the difficulty in emotion perception. It seems reasonable that both extremes of valence (highly positive and highly negative) might be linked to particularly distinctive emotional expressions (e.g., ecstasy or joy on the positive side, desperation or fury on the negative side) as compared to expressions around the middle values (e.g., ease, calmness), and this might result in reduced difficulty of emotion perception at the extremes of the valence dimension. A similar effect may exist for arousal (e.g., easier recognition of fury or sadness than of mild interest). To address these possible extremeness effects, we examined possible quadratic relationships of both valence and arousal with emotion perception difficulty.

## Hypotheses

Based on the summarized emotion recognition literature, we formulated the following hypotheses to examine person-specific (age and sex of observer and actor) and stimulus-specific (valence and arousal of the displayed expression) variables as predictors for subjective and objective difficulty in emotion perception:

**Hypothesis 1:** Age and sex of the actor have an effect on the objective and subjective difficulty experienced by the observer. We hypothesized that observers would experience more difficulty when presented with expressions from older actors (due to reduced muscle control) as compared to younger ones, and from male actors (due to lower expressiveness) in comparison to female actors.

**Hypothesis 2:** Age and sex of the observer have an effect on the observer's objective and subjective difficulty in emotion expression perception. We expected to observe difficulty following a quadratic function across age because of developmental and deterioration effects at the two ends of the age spectrum. We also expected that difficulty would be increased for male observers compared to female observers (due to the latter's advantage in emotion recognition).

**Hypothesis 3:** Valence and arousal of the stimuli, as subjectively perceived by a participant, have an effect on subjective and objective difficulty in emotion perception of these stimuli. We expected to see negative quadratic relationships for both valence and arousal in subjective and objective difficulty.

## Methods

### Participants

In total 658 volunteers took part in an online survey in German. The study was approved by the Ethics Committee at the Charité –Universitätsmedizin Berlin and was conducted according to the principles of the Declaration of Helsinki. All participants provided written informed consent for participation. Participants were included if they were either female or male native German speakers and not undergoing psychotherapy or taking psychoactive medication (to exclude clinical groups with possible emotion perception deficits, e.g., in bipolar disorder, depression, schizophrenia; [45,46]) at the time of the study. No upper age restrictions were included in the initial inclusion criteria, but few data sets from participants older than 60 were collected. To avoid spurious age effects due to the high leverage of data points from a small number of the oldest participants (oldest 79) in the planned regression models, in the final sample we included only participants who were 60 years old or younger (see S1 Fig in the S1 File for a histogram of age in the final sample). The participants were informed that their participation (and completion of the experiment) would allow them to enter a raffle with 50-euro prizes. After the data collection, 3 participants were randomly selected and awarded the prizes (via bank transfers).

**Exclusion criteria.** A number of pre-planned steps were taken to clean the data before the analysis due to the unsupervised nature of the study: It was administered via an online platform and some participants could potentially rush through it without paying attention to its content. Hence, participants were only included if they reported working video playback and if they passed a simple test of their reliability in emotion reporting, which consisted of watching a short video of an actor showing a prototypical expression of happiness and choosing the label "happiness" from a set of basic emotions and an "I do not know" option. The data sets were furthermore filtered based on improbable response times: on top of 4 seconds playback time of the video clips, we estimated a minimum of 2 seconds for a response on each scale (determined by visual inspection of the distribution of average times per page across all subjects), which resulted in filtering out participants completing valence-arousal rating pages in an average time lower than 10 seconds and emotion rating pages in an average time lower than 20 seconds.

Further, happiness is frequently used as a direct measure of positive affect (e.g., in the Positive And Negative Affect Schedule; [47]) and therefore happiness and valence ratings should exhibit a high correlation. This was confirmed in our data by visual inspection of the distribution of within-participant valence–happiness correlations which showed a clear division around 0.4. Thus, this value was used as an exclusion threshold and participants whose responses showed lower correlation were excluded. The mean correlation between happiness and valence was 0.87 before and 0.88 after this filtering step.

In total 217 participants were excluded and 441 remained (129 males). Mean age was 28.08 ±8.17 for women and 29.82±8.88 for men. German cultural background was reported by 423 participants. Asked about ethnicity, 380 participants reported white, 23 other, and 38 chose not to provide an answer to this question.

**Justification of the sample size.** The minimum sample size was set to 400 participants, which would ensure that on average every one of the 480 video clips used in the study (see

Materials) was rated by 10 participants (in the final sample the average number of raters was 11; min: 5, max: 18). We collected more participants initially because we had made an estimate of having to exclude approximately 30% due to the aforementioned exclusion criteria.

## Materials

The present study used a set of 480 four-second long videos with 12 actors (6 male, 6 female, age range: 21–64, mean age: 35.92) in which an actor's face, centrally presented on a grey background, changed from a neutral expression to display an emotion and then returned to the neutral expression. These videos were a part of a larger set produced for research purposes at the film studio of the Humboldt-Universität zu Berlin (for a detailed description, see [48]). The actors were provided with the target emotion label, situations in which it is likely to occur in real life, and physiological changes associated with it. The video clips were then validated by independent judges and yielded high emotion recognition rates and good believability. The emotional facial expressions in the subset used in this study spanned over 40 different emotion categories (Supplementary Material, S1 Table in S1 File), including Ekman's six basic emotions and 34 complex emotions. These categories were selected in previous research based on their frequency in everyday life and their even distribution across valence and arousal [49] and in the present study they assured a wide range of emotional expressions across the stimuli set. We included both basic and blended expressions in order to ensure that our results would generalize beyond the basic emotions. However, the original labels (i.e., the target emotion actors were instructed to express) was not used in this study because we were not interested in "correctness" of emotion recognition, but rather in the difficulty of emotion perception.

## Procedure

Testing was carried out in German on the soscisurvey.de platform [50]. Each participant rated 12 videos randomly chosen from the pool of 480 videos. First, participants rated valence and arousal on scales anchored by pictures of the respective Self-Assessment-Manikins [51]. Second, participants provided ratings on the Basic Emotions and Interest (BEI) scales: happiness, sadness, fear, anger, surprise, disgust, interest. These scales were chosen to represent readily understandable emotions spanning positive (e.g., happiness), negative (e.g., disgust), and neutral/ambiguous (e.g., surprise) valence, higher (e.g., anger) and lower (e.g., sadness) levels of arousal, and higher (e.g., interest) and lower (e.g., fear) complexity. Together, we judged these scales to be good candidates for dimensions which could approximate the larger emotion space. Finally, participants rated the subjective difficulty of making the BEI ratings. All ratings were given on continuous visual sliding scales ranging from "not X at all" to "completely X", where X was the rated emotion word. Responses closer to "completely X" were reflecting higher perceived intensity of emotion X in the rated video. For the difficulty rating the extreme ends of the scale were "very easy" to "very difficult". All rating responses were encoded between 1 and 101.

## Measures

The dependent variables were self-rated difficulty (SRD) and objective difficulty (OD). The SRD measure was the subjective difficulty of making the BEI ratings provided by the participants. The OD measure was calculated as the Euclidean distance from an observer's rating to the consensus of the population on a given video. The population consensus was calculated as the centroid in the seven-dimensional BEI rating space of each video. This distance therefore reflects how similar an individual observer's perception of a particular video was to the average perception of that video. Because according to the literature discussed above (see 1.2.4) it

seemed likely that demographic-based subgroups in the population have different consensus (e.g., females and males could exhibit in-group agreement but between-group differences; see 1.1.3), we had to first test whether observer sex and age predicted significantly the BEI ratings (we call this test a consensus check). For that, the BEI ratings served as dependent variables in models with observer sex and age as predictors.

## Data analysis

**Model structure.** All the models were fit with random intercepts for observers and videos. In cases where including both random effects overly reduced power and interfered with estimation of fixed effects, one or the other was omitted, as described below. Additionally, we tested whether random slopes for videos or actors nested in observers would account for stimulus-related inter-subject variance in null models (i.e., without fixed effects; see Supplementary Material, S7 Table **in S1 File**). However, no full model (testing fixed effects) could be converged with this large random effect structure. Thus, here we report models with random intercepts for observers and videos. Across all models, *i* is an index iterating over participants, *j* is an index iterating over actors and *k* is an index iterating over videos.

**Consensus check.** Prior to the hypotheses testing, we built a model for each BEI item with the rating as the dependent variable and observer age and observer sex as independent variables to check for their potential systematic effects, which was important for the calculation of the OD (see 2.4.). Eq 1 shows this for the happiness rating as an example. The happiness rating for a video *k* rated by observer *i* is estimated by an intercept $\beta_0$, equal to the mean happiness rating across videos and observers, plus a term $\beta_1$ for observer sex and a term $\beta_2$ for observer age. Additionally, the model contains a random effect $\beta_{video_k}$, which adjusts the intercept $\beta_0$ for each video *k* and therefore accounts for the repeated occurrence of the videos.

$$Happiness\ rating_{ik} = \beta_0 + \beta_{video_k} + \beta_1 * (observer\ sex_i) + \beta_2 * (observer\ age_i) + \epsilon_{ik} \qquad (1)$$

**Confirmatory analyses.** Hypothesis 1 was examined with two mixed-effects models (Eqs 2 and 3), one for each difficulty measure (OD and SRD) as the dependent variable. Fixed effect variables were actor sex and actor age. A random intercept for observer was specified $\left(\beta_{observer_i}\right)$ to account for the repeated measurements of each observer *i*. Since the variables of interest were actor-related and each video stimulus was nested within actor, no random effect for the multiple appearances of video stimuli was included in this model. We instead assume that the fixed effects of actor sex and actor age already account for the common variance of a given video.

$$OD_{ij} = \beta_0 + \beta_{observer_i} + \beta_1 * \left(actor\ sex_j\right) + \beta_2 * \left(actor\ age_j\right) + \epsilon_{ij} \qquad (2)$$

$$SRD_{ij} = \beta_0 + \beta_{observer_i} + \beta_1 * \left(actor\ sex_j\right) + \beta_2 * \left(actor\ age_j\right) + \epsilon_{ij} \qquad (3)$$

In a similar way, two mixed-effects models (Eqs 4 and 5) were specified for hypothesis 2 with observer sex, observer age and squared observer age as fixed effects, and a random intercept term for the rated video $\left(\beta_{video_k}\right)$. The simple and the squared observer age variables were mean-centered to reduce collinearity of these predictors. Again, a random intercept term for observer could not be easily specified due to fitting difficulties; however, we continue using observer age and sex, both nested under that variable. While this way of modeling does leave

repeated measurements of the observers in the data, their influence should be accounted for by the fixed effects.

$$OD_{ik} = \beta_0 + \beta_{video_k} + \beta_1 * (observer\ sex_i) + \beta_2 * (observer\ age_i) + \beta_3 * (observer\ age_i^2) + \epsilon_{ik} \quad (4)$$

$$SRD_{ik} = \beta_0 + \beta_{video_k} + \beta_1 * (observer\ sex_i) + \beta_2 * (observer\ age_i) + \beta_3 * (observer\ age_i^2) + \epsilon_{ik} \quad (5)$$

Hypothesis 3 was examined with two mixed effects models using valence and arousal as linear and squared independent variables (Eqs 6 and 7). Random intercepts for video $\left(\beta_{video_k}\right)$ and observer $\left(\beta_{observer_i}\right)$ were included in the model to account for the repeated measurements of observers and the repeated occurrence of rated videos. Because the squared terms (e.g., valence$^2$) can be seen as interaction terms (valence x valence), all constitutive terms (e.g., valence) should be included in the models, but not interpreted [52]. For this reason, we first built the models without the squared terms, and then fitted models with additional squared terms. Valence and arousal entered the model as Z-standardized variables to reduce collinearity between the linear and squared terms and to ensure comparability of estimated coefficients. An additional multiple testing correction was applied for the subsequent inclusion and testing of the squared term (see next section).

$$OD_{ik} = \beta_0 + \beta_{observer_i} + \beta_{video_k} + \beta_1 * (valence_i) + \beta_2 * (valence_i^2) + \beta_3 * (arousal_i) + \beta_4 * (arousal_i^2)$$
$$+ \epsilon_{ik} \quad (6)$$

$$SRD_{ik} = \beta_0 + \beta_{observer_i} + \beta_{video_k} + \beta_1 * (valence_i) + \beta_2 * (valence_i^2) + \beta_3 * (arousal_i) + \beta_4 * (arousal_i^2)$$
$$+ \epsilon_{ik} \quad (7)$$

**Exploratory analyses.** Additional exploratory models (Supplementary Material, Equations in S1 File) were run to investigate results from the previous models.

**Multiple testing correction.** Multiple testing correction was applied to ensure a consistent alpha level of 0.05. Models built on Eq 1 were subjected to a 7-fold multiple testing correction to account for the repeated testing (separately for each rating dimensions) of the assumption that ratings were correlated with observer sex and/or age. Models of hypotheses 1 and 2 were subjected to 2-fold multiple testing correction each to account for the common structure of OD and SRD models and the correlation in the outcome values. Models of hypothesis 3 were subjected to a 4-fold testing correction for the aforementioned reason of common structure and because models only with linear predictors were fitted prior to full models (including linear and squared terms). For all corrections the Bonferroni-Holm method was used.

**Software.** All analyses were conducted in RStudio [53] using R 3.3.3 [54] under Windows 7. Mixed effects models were built using the "lme4" package [55]. Ordinary linear regression was conducted with the functionality of the base R package. P-values for mixed effects models were provided by "lmerTest" package version 2.0–30 [56]. Model output was arranged in tables using the R "stargazer" package [57].

**Feature importance.** Since our aim was to understand which person- and stimulus-specific factors predict subjective and objective difficulty in emotion perception, we needed a way to compare the predictive strength of the variables of interest in the data. For this, we calculated feature importance. Feature importance estimates the relative impact of a given predictor on the ability of a model to predict the target variable, in our case: difficulty measures. Here, feature importance was calculated by the increase in node purity measured in Gini in a

random forest model similar to work by Brick, Koffer, Gerstorf, & Ram [58] using the "caret" package [59].

## Results

### Consensus check

For the calculation of our objective difficulty measure (OD) it was important to know if observer sex and age significantly influenced their BEI ratings (see 2.4.).

The analysis revealed no significant effect of observer's age. However, as expected based on the previous literature, statistical tests revealed an influence of observer's sex on the rating dimensions. A significant influence of observer sex was found for all emotion dimensions except "happy" (Supplementary Material, S2 Table in S1 File). The coefficients of the observer sex term are negative, indicating that women rate these dimensions systematically lower than men do. Therefore, in all the planned analyses including observer's sex as a predictor we use the OD measure calculated as the distance from a separate centroid for each sex (to reflect the sex-specific consensus in the BEI ratings). In the process of this analysis, 171 video clips were left with ratings by less than 3 male observers; These videos were excluded.

### Confirmatory analyses

Table 1 shows models examining the relationship between the two difficulty measures and actor sex and age according to hypothesis 1. For SRD a significant positive effect was found for actor age, indicating that with actor age the perceived difficulty for observers increases. For the OD measure a positive effect was found for actor sex and age, showing that OD increases with increasing actor age and is higher for female actors.

Hypothesis 2 was tested with the models depicted in Table 2. A positive association of SRD with female observer sex and a negative quadratic effect for observer age (Table 2(1)) were

**Table 1. Mixed effects models for subjective and objective difficulty predicted by actor sex and age and a random effect for observer.** Exact p values are shown in the brackets next to the unstandardized estimates. 95% Confidence Intervals are shown in square brackets.

| | Dependent variable: | | | |
|---|---|---|---|---|
| | Self-rated Difficulty Unstandardized Standardized | | Objective Difficulty Unstandardized Standardized | |
| Fixed effects | (1) | (2) | (3) | (4) |
| Actor female | 0.36 (.57) | 0.01 | 1.84 (.005)** | 0.09* |
| | [-0.89, 1.61] | [-0.03, 0.06] | [0.57, 3.11] | [0.03, 0.16] |
| Actor Age | 0.06 (.016)* | 0.03* | 0.06 (.023)* | 0.04* |
| | [0.01, 0.11] | [0.01, 0.05] | [0.01, 0.11] | [0.01, 0.07] |
| Intercept | 38.03 ($<$ .001) | -0.01 | 49.54 ($<$ .001) | -0.04 |
| | [35.66, 40.41] | [-0.07, 0.05] | [47.47, 51.62] | [-0.10, 0.01] |
| Random effects | SD | SD | SD | SD |
| Observer | 14.37 | 0.54 | 5.95 | 0.30 |
| Residual | 22.33 | 0.84 | 18.73 | 0.95 |
| Observations | 5,292 | 5,292 | 3,537 | 3,537 |
| Log Likelihood | -24,338.78 | -6,993.32 | -15,513.14 | -4,981.63 |
| Akaike Inf. Crit. | 48,687.57 | 13,996.65 | 31,036.29 | 9,973.25 |
| Bayesian Inf. Crit. | 48,720.44 | 14,029.52 | 31,067.14 | 10,004.11 |
| Marginal $R^2$ | 0.001 | 0.001 | 0.004 | 0.004 |
| Conditional $R^2$ | 0.294 | 0.294 | 0.095 | 0.095 |

Note: *p$<$0.05.

**Table 2. Mixed effects models for self-rated difficulty and objective difficulty predicted by observer sex and age and a random effect for video.** Exact p values are shown in the brackets next to the unstandardized estimates. 95% Confidence Intervals are shown in square brackets.

| | Dependent variable: | | | |
| --- | --- | --- | --- | --- |
| | Self-rated Difficulty Unstandardized Standardized | | Objective Difficulty Unstandardized Standardized | |
| Fixed effects | (1) | (2) | (3) | (4) |
| Observer female | 3.51 (< .001)*** | 0.13*** | 3.58 (< .001)*** | 0.18*** |
| | [1.96, 5.05] | [0.07, 0.19] | [2.33, 4.83] | [0.12, 0.25] |
| Observer Age | 0.14 (.025) | 0.01 | 0.05 (.406) | 0.002 |
| (mean centered) | [0.02, 0.27] | [0.001, 0.01] | [-0.06, 0.15] | [-0.003, 0.01] |
| Observer Age$^2$ | -0.02 (< .001)*** | -0.001*** | 0.002 (.476) | 0.0001 |
| (mean centered) | [-0.02, -0.01] | [-0.001, -0.0003] | [-0.004, 0.01] | [-0.0002, 0.0005] |
| Intercept | 38.94 (< .001) | -0.06 | 49.91 (< .001) | -0.13 |
| | [37.402, 40.472] | [-0.11, 0.002] | [48.51, 51.32] | [-0.20, -0.06] |
| Random effects | SD | SD | SD | SD |
| Video | 7.43 | 0.28 | 8.02 | 0.41 |
| Residual | 25.42 | 0.96 | 17.90 | 0.91 |
| Observations | 5,292 | 5,292 | 3,537 | 3,537 |
| Log Likelihood | -24,793.15 | -7,453.49 | -15,409.75 | -4,883.73 |
| Akaike Inf. Crit. | 49,598.29 | 14,918.99 | 30,831.49 | 9,779.47 |
| Bayesian Inf. Crit. | 49,637.73 | 14,958.43 | 30,868.52 | 9,816.49 |
| Marginal R$^2$ | 0.01 | 0.006 | 0.01 | 0.01 |
| Conditional R$^2$ | 0.08 | 0.084 | 0.17 | 0.17 |

Note: ***p<0.001.

found. The latter indicates that the youngest and oldest observers in the sample tended to rate items as less difficult than the middle age group. OD likewise showed a positive association with female observer sex but no significant association with observer age (Table 2(3)).

Table 3 shows models estimating the influence of valence and arousal ratings on SRD and OD according to hypothesis 3. For SRD (Table 3(1)) negative squared effects of valence and arousal were observed. This means that on average stimuli rated on the low or high ends of the valence or arousal scales are reported as easier than stimuli with ratings towards the middle on these scales. Comparison of the coefficients for the squared valence and arousal terms indicate a steeper slope for valence than for arousal. In the OD model (Table 3(3)) a negative coefficient for the squared valence term was found, which indicates lower OD for extreme ratings on the valence scale in contrast to higher OD for ratings towards the center of the scale. However, the squared arousal predictor has a positive coefficient. This indicates a lower OD for stimuli rated as medium-level in arousal as opposed to stimuli rated towards the extreme ends of the scale.

## Exploratory analyses

Exploratory analyses were performed to complement the confirmatory results and provide further grounds for interpretation of the data. Because of the exploratory nature no p-values are provided.

**Difficulty measures in relation to valence and arousal.** The relationship of both difficulty measures with valence and arousal was plotted for mean values for each video clip (Fig 1) which provides a simple way of visualization that the mixed effects models do not allow for easily. We fit simple linear models to all four relationships (Supplementary Material, Eqs 12–15). Models including both a linear and a squared term for valence showed a good fit, indicated by adjusted

**Table 3. Mixed effects models for self-rated and objective difficulty predicted by valence and arousal ratings and random effects for video and observer.** Exact p values are shown in the brackets next to the unstandardized estimates. 95% Confidence Intervals are shown in square brackets.

| | Dependent variable: | | | |
| --- | --- | --- | --- | --- |
| | Self-rated Difficulty<br>Unstandardized Standardized | | Objective Difficulty<br>Unstandardized Standardized | |
| Fixed effects | (1) | (2) | (3) | (4) |
| Valence (standardized) | 2.44 (< .001)*** | 0.09*** | -6.04 (< .001)*** | -0.30*** |
| | [1.67, 3.20] | [0.06, 0.12] | [-6.73, -5.36] | [-0.33, -0.26] |
| Valence (standardized)$^2$ | -9.29 (< .001)*** | -0.35*** | -1.48 (< .001)*** | -0.07*** |
| | [-9.99, -8.58] | [-0.38, -0.32] | [-2.09, -0.88] | [-0.10, -0.04] |
| Arousal (standardized) | 0.36 (.271) | 0.01 | 0.71 (.014)* | 0.03 |
| | [-0.28, 1.01] | [-0.01, 0.04] | [0.14, 1.27] | [0.01, 0.06] |
| Arousal (standardized)$^2$ | -2.30 (< .001)*** | -0.09*** | 1.14 (< .001)*** | 0.06*** |
| | [-2.97, -1.62] | [-0.11, -0.06] | [0.57, 1.72] | [0.03, 0.08] |
| Intercept | 51.98 (< .001) | 0.44 | 55.75 (< .001) | 0.01 |
| | [50.28, 53.68] | [0.37, 0.50] | [54.56, 56.94] | [-0.05, 0.07] |
| Random effects | SD | SD | SD | SD |
| Video | 4.45 | 0.17 | 5.18 | 0.25 |
| Observer | 13.96 | 0.53 | 7.22 | 0.35 |
| Residual | 19.84 | 0.75 | 16.83 | 0.82 |
| Observations | 5,292 | 5,292 | 5,292 | 5,292 |
| Log Likelihood | -23,842.71 | -6,506.34 | -22,868.14 | -6,913.66 |
| Akaike Inf. Crit. | 47,701.43 | 13,028.68 | 45,752.28 | 13,843.31 |
| Bayesian Inf. Crit. | 47,754.02 | 13,081.27 | 45,804.88 | 13,895.90 |
| Marginal R$^2$ | 0.13 | 0.13 | 0.12 | 0.12 |
| Conditional R$^2$ | 0.44 | 0.44 | 0.32 | 0.32 |

Note: ***p<0.001.

$R^2$ values of 0.37 and 0.51 for dependent variables SRD and OD respectively (Fig 1A and 1B). We also fit models containing a linear and a squared term for arousal and SRD and OD respectively as the dependent variable to the data. However, the fit was poor as indicated by the very low adjusted $R^2$ values of 0.05 and 0.01. Graphs c and d in Fig 1 show these much less obvious patterns in the data.

**Feature importance comparisons of all variables.** Feature importance scores are difficult to interpret in raw form but can be used as an intuitive guide to the relative impact of different predictors. Here, we present importance relative to the most important predictor (Fig 2), such that a score of 1 indicates the most powerful predictor, and a score of .5 indicates that a given predictor carries half the predictive power of that most powerful predictor. Feature importance calculations show that the valence rating is the strongest predictor for SRD (Fig 2A), followed by the happy rating with 74% of the importance of valence. Most other predictors improve accuracy by 56% (interested rating) to 42% (disgusted rating) as much as valence does. The exceptions are the predictors actor age (30%), observer sex (10%) and actor sex (7%), which seem to be of rather low importance for the SRD prediction. For the OD measure the best predictor is the happy rating, followed closely by the interest rating (93% of happy rating) and the valence rating (79% of happy rating). Most other predictors improve accuracy by about 62% (fearful rating) to 50% (arousal rating) of the happy rating predictor. However, the predictors of lowest importance are all person-specific variables: observer age (36%), actor age (31%), actor sex and observer sex (6%).

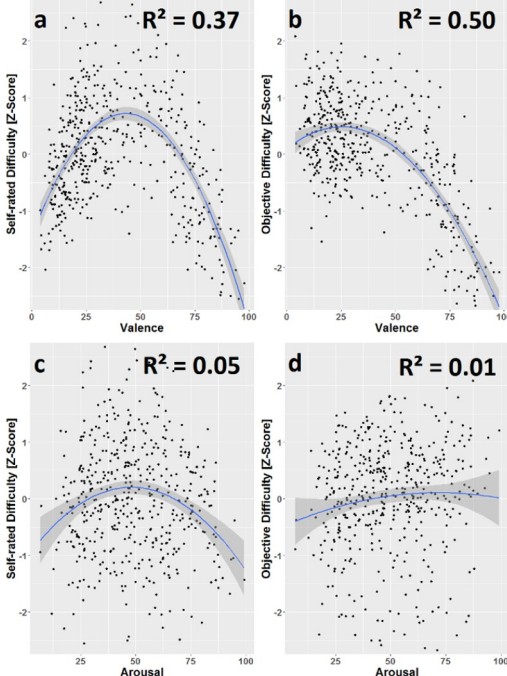

**Fig 1. Plots visualizing the relationship of valence and arousal with both difficulty measures.** Data points are averages over videos. The ribbon depicts the 95% confidence interval. A clear negative curvilinear relationship can be seen between valence and SRD (a). A similar relationship *exists* for valence and OD (b), although, the curve is less symmetrical with the high valence region featuring the lowest OD values. For the arousal measure the data does not exhibit such a visually clear relationship with the OD or SRD measure; this is also indicated by the low adjusted $R^2$ values (c,d).

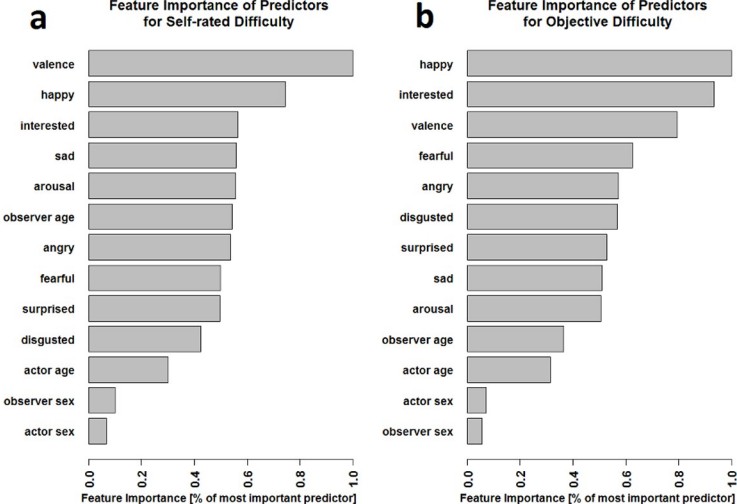

**Fig 2.** Feature importance ranking for the prediction of SRD (a) and OD (b). The importance is expressed as a proportion of the importance of the strongest predictor and displayed in decreasing order from top to bottom. For the SRD measure the valence and happy ratings show a high importance, while all of the observer- and actor-specific variables except observer age show a particular low importance. For the OD measure the happy rating is the most important predictor with the interest and valence ratings following up close in importance.

## Discussion

The aim of this study was to investigate which person-specific (age and sex of actors and observers) and stimuli-specific variables (valence and arousal) constitute the subjective and objective difficulty of the perception of emotional facial expressions.

### Age and sex of actor

We predicted that both actor's sex and age influence subjective and objective difficulty in emotion perception. In line with hypothesis 1 and previous research [29], we found that emotions displayed by older actors were rated on average as more difficult to judge (subjective difficulty) and that ratings for older actors were more dispersed (individual ratings farther away from the group's consensus), which indicates more disagreement between observers (objective difficulty). On the other hand, an effect for actor's sex was only found in the model predicting objective difficulty and, surprisingly, it yielded higher difficulty linked to female actors. While the literature describes greater facial expressivity in women [31,32], it is unclear how this relates to the observers' difficulty in emotion perception. In a categorical emotions framework, greater expressivity should result in stronger expressions of the primary emotion, and therefore a clearer signal that can be more easily interpreted. However, in a multidimensional emotion perception framework greater expressivity may result in stronger expression of all the emotions (primary and secondary), which could render the signal more difficult or ambiguous to decode. Results of the additional exploratory models (Supplementary Material, Eqs 8 and 9) are in fact consistent with this possibility; They show that on average 23.93 more rating points (Supplementary Material, Table S3.1 in S1 File) were spent describing the expressions of female actors and that the standard deviation across ratings was 2.23 units higher for female actors (Supplementary Material, Table S3.2 in S1 File). This indicates that female actors were perceived by the observers as expressing a stronger but also more complex emotional signal.

### Age and sex of observer

According to hypothesis 2, the observer's age and sex should influence both subjective and objective difficulty. We reasoned that difficulty should follow a quadratic function to account for developmental [37–39] and aging effects [40,41] on both sides of the age spectrum. However, we found no quadratic effect of age on the objective difficulty and a *negative* quadratic effect for the subjective difficulty. Younger and older observers within the age range of our sample rated themselves as having on average less difficulty than the middle-aged observers, while in the objective difficulty model there was no effect for the same age coefficients. Although a difference in significance patterns between the models does not mean a significant difference between the results, this is consistent with the idea that younger and older people might overestimate their perceptual abilities, which has been discussed before in metacognition literature [60].

Female observers reported having more subjective difficulty in providing the Basic Emotions and Interest (BEI) ratings and also exhibited higher objective difficulty. The former falls in line with studies showing that males exhibit overconfidence, i.e., more confidence than justified by their own abilities, in stock trading [61], test taking [62] and the usage of technology [63]. It is therefore likely that the same effect carries over to the domain of emotion recognition, which may underlie the differences between men and women in our study. By contrast, it is unexpected that women also scored lower on the objective difficulty measure, since based on the literature pointing at the advantage of females in emotion *recognition* [42] we expected also less difficulties in this group in emotion *perception*. One explanation might be that female observers see more subtle signals in facial expressions (which have a greater role in a

multidimensional perception framework than in recognition tasks) but that these perceived secondary signals may not be generally agreed upon in this group, thus resulting in more deviation from the general consensus (i.e., higher objective difficulty).

To further this observation, we conducted exploratory analyses (Supplementary Material, Eqs 9 and 10), which revealed that the standard deviation across rating scales was more than one unit higher for female than male observers (Supplementary material, Table S4.2 in S1 File). This hints at greater disagreement among females. Moreover, women assigned on average 2.57 more rating points on the BEI scales for the basic emotions (angry, happy, sad, disgusted, surprised, interested, fearful), which is the only subset of our stimuli where the "target" emotion's label (according to the stimuli set) can be found in the rating scales (Supplementary Material, Table S4.3 in S1 File). This is in line with the literature that suggests a subtle advantage of women for emotion *recognition* [3,4]. Also, this rating pattern of greater mean response to target emotions with a higher standard deviation in females has been previously reported [11].

Therefore, our results in part reproduce previous findings (greater target emotion attribution for women) and in part introduce novel and unexpected findings (greater objective difficulty for women). Since the higher objective difficulty in females was manifested thanks to the multidimensional emotion framework introduced in this study, this should be an indicator that the traditional emotion recognition paradigms might not be sufficient for the detection of some of the perceptual differences between the sexes. Thus, future research should also include novel approaches (like the multidimensional emotion framework) to address the bigger picture.

## Stimulus valence and arousal

We predicted an influence of valence and arousal ratings on both difficulty measures. For the subjective difficulty measure as the dependent variable, both valence and arousal exhibited a negative quadratic relationship, indicating that stimuli rated on the extreme ends of the valence and arousal scales were also perceived as less difficult. The standardized coefficients revealed that the influence of valence is almost four times stronger than that of arousal (Table 3(2)). In the objective difficulty model, valence showed again a negative quadratic relationship. Arousal, on the other hand, followed a *positive* quadratic function, meaning that high and low arousal expressions led to higher values on the objective difficulty measure, i.e., more disagreement between observers (Table 3(4)). However, the magnitude of the quadratic effects for arousal and valence are similar.

In related research, we found that arousal ratings are correlated with the displacement from a neutral face (e.g., distance between the placement of the corners of the lips in a happy expression and in a neutral expression; [64]). It has also been shown before that facial expressions of high-intensity emotional states are often misclassified in terms of valence (an expression arising from a positive experience perceived as one of negative valence and vice-versa) in the absence of further information, such as body posture [65,66]. What we observe in the model for the objective difficulty measure could be similar: There might be some level of arousal above which discrimination of facial expressions worsens. On the other hand, too little deviation of the face from a neutral expression may also result in low arousal ratings and potentially also provide an observer with little information on the emotional meaning, thus making it difficult to recognize the displayed expression.

Further data exploration on video clip averages (Fig 1) also revealed negative quadratic relationships between valence, arousal, and both difficulty measures, except for the positive quadratic relationship of arousal and objective difficulty (Fig 1D). Valence seemed to be more

predictive in terms of explained variance of self-reported (37%) and objective difficulty (51%) than arousal (5% and <1%, respectively). Importantly, these results show that valence is the predominant predictor for both difficulty measures.

The asymmetry of the curves in Fig 1A and 1B shows that high valence stimuli are the least difficult, even less than stimuli from the very low end of the valence spectrum. It is known that within basic emotion paradigms stimuli from the happiness category show the highest recognition rates and often produce ceiling effects [3,67,68]. One could argue that this pattern might be an artefact of the BEI dimensions, where happiness is the only definite positive emotion (in contrast to angry, sad, fearful, and disgusted being negative, and interested and surprised being neutral/ambiguous). In this limited set, the range of choice for positive emotions is decreased which could be reflected in the observers' difficulty perception (measured as self-reported difficulty) and also in the dispersion of overall ratings (objective difficulty). We believe, however, that the shape of the curves reflects the true relationship of valence and difficulty, because it is highly stable across various subsets of the data. For example, if the data are split into the basic emotions and the remaining "complex" emotions (thus removing pure "happiness" from the latter), the observed effect of valence holds in both subsets. Moreover, this effect persists even when only looking at stimuli of a single emotion category (as defined in the used stimuli set). For example, for the "surprised" category, which is not easily classified as either positive or negative, the individual stimuli are still distributed along the same curves. This strongly indicates that this effect is neither driven by the influence of individual stimulus categories, nor by the choice of dimensions in the BEI ratings, nor an interaction of those. It seems to rather reflect an underlying phenomenon. Further research is needed to understand the meaning and causes of this effect in more detail, but it indicates an interesting sort of non-linearity in the detection of emotions with extreme valence.

## Effect sizes and feature importance of predictors

Marginal $R^2$ values express the variance explained by the fixed effects. The marginal $R^2$ values in all models testing person-specific variables for their influence on both difficulty measures (Tables 1 and 2) were close to zero and thus represent very small effects [69]. Conversely, marginal $R^2$ values for models with valence and arousal (Table 3) were in the range of moderate sized effects. Together, these imply a limited influence of person-specific factors and larger effects of stimulus-specific factors on emotion perception.

Similarly, the feature importance analyses (Fig 2) showed that the best predictors for subjective difficulty were ratings of overall valence and the scale "happy", whereas for objective difficulty, overall valence, "happy", and "interested" dominated, emphasizing once again the importance of a general pleasure dimension for emotion perception. For both difficulty measures, the person-specific measures showed low importance with the exception of observer age in the subjective difficulty model. This implies that the impact of these person-specific features may be swamped by the difficulty differences between emotions, but that observer age may still have an important impact on how difficult emotion ratings are perceived.

The rating of interest was the third strongest predictor for subjective and second strongest predictor for objective difficulty. The interest dimension may be selectively indicating expressions of "social emotions", i.e., emotions that are directed at another person, in contrast to those which do not require the presence of an interaction partner. Social emotions are usually more subtle in their display and more dependent on context, which may make them more difficult to evaluate. Future work should investigate whether social emotions are in general more difficult and whether they can be separated by ratings of interest from non-social emotions to confirm this. Another explanation for the high importance of the interest dimension might be

that interest ratings also differentiate between negative and positive emotions and thus act as a proxy for emotion valence. In fact, in our data valence and interest ratings are moderately (r = .40) and strongly (r = .63) correlated on the individual rating and video level, respectively (Supplementary Material, S5 and S6 Tables in S1 File).

## Limitations and future directions

One potential limitation of this study is the sample size and the ratio of both sexes in relation to the calculation of the objective difficulty measure. We calculated the ground truth for each video clip from our sample of observers by averaging all ratings for a clip within groups of males and females, which demonstrates how the difficulty of emotion perception can be captured in a multidimensional emotion framework and without assigning an *a priori* ground truth. Nonetheless, the precision of this estimate is dependent on the sample size. Due to our study design each individual video clip was only rated by a small group of the total observer pool: On average 11 observers rated a video clip, which was further broken down for some analyses into male and female observers. However, because we were only interested in effects across videos, a low observer count on individual clips should not systematically bias our results. In fact, the decision to have observers randomly distributed over many video clips instead of having all observers rate the same few video clips adds to the generalizability of our results as they were observed over a wide array of emotions and actors. Thus, we opted for maximizing external validity of our design and results. Nevertheless, one could argue that because each participant rated a subset of the stimuli, this could introduce inter-subject bias in the data. While we cannot eliminate this possibility entirely, we argue that our design is suited for maximizing the external validity of the study and that such a bias is likely limited and accounted for in our results. Specifically, we considered maximum random effects structure in our models (including random slopes nested within subjects). However, these models did not converge. A likely reason for this is that, as we assume and explain for each hypothesis, the fixed effects already account for that variance (i.e., inter-subject bias), thus rendering our models suitable for testing the stated hypotheses (for a detailed comment, see Supplementary Material).

Further, one could argue that the concept of difficulty in our study is task-dependent due to the limited number of rating dimensions and as such cannot be generalized beyond the stimuli set used here. In this view, the difficulty experienced by the participants in the task is a consequence of having to "deconstruct" a given (sometimes complex) emotion expression into maximally seven proposed dimensions (the BEI ratings).

Although we cannot rule out for certain that with the chosen dimensions we covered only a limited part of the emotion space, it is likely that this part is rather broad as it has been previously shown that merely four dimensions (valence, dominance, arousal, unpredictability) can explain 75% of the variance in a space of 144 categories [9]. As long as the seven BEI ratings are not all limited to a particular niche part of the emotion space, which seems unlikely, our results should generalize over different stimuli sets. Moreover, even if some part of the experienced difficulty could be explained by the characteristics of the task, it is important to note that it was the same across participants: All were equally likely to be exposed to potentially less (basic) or more difficult (complex) emotions. Since the variables of interest in the study were person- and stimulus-specific *across* different emotions (more and less difficult), our results and conclusions are informative regardless of whether the task itself could add to the perceived difficulty in emotion perception.

Another possible limitation is the use of only one positive rating dimension–happiness. This could potentially influence the ratings of positive videos, which would all likely include

high-intensity choices on the happiness scale. By contrast, the negative videos would be rated higher on the remaining non-positive six dimensions. Although it is true that our BEI ratings include more of obviously negative (sadness, anger, fear, disgust) than positive (happiness) emotions, there are two emotions with non-obvious valence (surprise, interest). In fact, the positive videos in this study were rated as combinations of happiness, surprise, and interest in various intensities. Moreover, even if the unbalanced dimension set in terms of valence would have an effect on the ratings between positive and negative videos, our analyses of the effects of person-specific predictors on emotion perception difficulty would not be influenced by this, as the stimuli of different valence were equally distributed across participants. Finally, the stimulus-specific analyses (of perceived valence and arousal), though exposed to this potential limitation, showed a largely resisting pattern of results across different subsets of the data (as explained in 4.3.). Together, these suggest that the observed effects in this study are not substantially influenced by the limited number of positive dimensions in the ratings. Nonetheless, future research should include more of clearly positive emotions as rating dimensions to reject the possibility that the chosen set could considerably affect the observed results.

## Conclusions

The present study employed a novel emotion perception paradigm and dynamic facial expression stimuli. This allowed the observers to rate stimuli on continuous dimensions rather than imposing discrete choices. With this paradigm we investigated the effects of stimuli- and person-specific variables on subjective and objective difficulty of emotion perception.

We believe that our study is the first to examine the influence of valence and arousal on the difficulty of facial expression perception. We showed negative quadratic relationships of valence and arousal with subjective difficulty, and valence with objective difficulty, as well as a positive quadratic relationship between arousal and objective difficulty. In these models, valence had a stronger effect than arousal on subjective difficulty. Further exploratory analyses gave strong evidence for a higher predictive importance of valence for both difficulty measures in contrast to arousal and person-specific predictors. The predominant role of valence for the difficulty of emotion recognition falls in line with functional accounts of emotions.

Consistent with the literature on emotion recognition, increased actor's age was linked to higher subjective and objective difficulty in emotion perception. The present study also provides evidence to suggest that certain groups overestimate their emotion recognition capabilities (as reflected in discrepancies between subjective and objective difficulty), in particular men and adults of low and high age, which is consistent with the confidence, development, and aging literature.

Taken together, these results fall in line with the functional account of emotion [70], which describes emotions as signals that carry survival-relevant information. In this view, emotional displays can be extreme in terms of movement and therefore might be rated high in arousal, but if no clear value judgment can be made, an emotional expression would remain difficult to decode. Thus, the high predictability of the difficulty of emotional perception by the valence dimension alone might be because emotions are in essence valence signals and humans are inherently tuned to them.

While most of our analyses reflected known effects supported by the literature, we also presented novel results. In particular, the increased objective difficulty for female actors and observers stands in contrast to the so-far reported findings and was possible to be revealed due to the use of multidimensional emotion framework in this study. This highlights the need to complement traditional emotion recognition paradigms with novel designs, like the one presented here, to capture a broader view of human emotion perception.

## Supporting information

**S1 File. Supplementary materials.**
(DOCX)

## Author Contributions

**Conceptualization:** Jan N. Schneider, Anne Weigand, Isabel Dziobek, Timothy R. Brick.

**Data curation:** Jan N. Schneider.

**Formal analysis:** Jan N. Schneider.

**Funding acquisition:** Isabel Dziobek.

**Investigation:** Jan N. Schneider.

**Methodology:** Jan N. Schneider, Anne Weigand, Isabel Dziobek, Timothy R. Brick.

**Project administration:** Jan N. Schneider.

**Resources:** Isabel Dziobek.

**Software:** Jan N. Schneider.

**Supervision:** Isabel Dziobek, Timothy R. Brick.

**Visualization:** Jan N. Schneider, Magdalena Matyjek.

**Writing – original draft:** Jan N. Schneider.

**Writing – review & editing:** Jan N. Schneider, Magdalena Matyjek, Anne Weigand, Isabel Dziobek, Timothy R. Brick.

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
