## [Decision Letter · Decision Letter 0]

1 Jul 2021

PONE-D-21-12940

Subjective and Objective Difficulty of Emotional Facial Expression Perception from Dynamic Stimuli

PLOS ONE

Dear Dr. Matyjek,

Thank you for submitting your manuscript to PLOS ONE. After careful consideration, we feel that it has merit but does not fully meet PLOS ONE’s publication criteria as it currently stands. Therefore, we invite you to submit a revised version of the manuscript that addresses the points raised during the review process.

 We have now reports from 2 experts in the field. Although Reviewer 1is more posituive, REviewer 2 requires some cha nges and I completely share this opinion. Please carefully address all points raised on the point-by-point basis in your rebuttal letter to me.

We look forward to receiving your revised manuscript.

Kind regards,

Marina A. Pavlova, PhD

Academic Editor

PLOS ONE

Journal Requirements:

3. We noted in your submission details that a portion of your manuscript may have been presented or published elsewhere. 

"This manuscript is based on and similar to a chapter in J. N. S.’ doctoral thesis available at: " ext-link-type="uri" xlink:type="simple">https://publishup.uni-potsdam.de/opus4-ubp/frontdoor/index/index/searchtype/collection/id/17101/start/1/rows/10/has_fulltextfq/true/docId/45927"

Please clarify whether this publication was peer-reviewed and formally published. If this work was previously peer-reviewed and published, in the cover letter please provide the reason that this work does not constitute dual publication and should be included in the current manuscript.

Reviewers' comments:

Reviewer's Responses to Questions

**Comments to the Author**

1. Is the manuscript technically sound, and do the data support the conclusions?

Reviewer #1: Yes

Reviewer #2: No

2. Has the statistical analysis been performed appropriately and rigorously? 

Reviewer #1: Yes

Reviewer #2: Yes

3. Have the authors made all data underlying the findings in their manuscript fully available?

Reviewer #1: Yes

Reviewer #2: No

4. Is the manuscript presented in an intelligible fashion and written in standard English?

Reviewer #1: Yes

Reviewer #2: No

5. Review Comments to the Author

Reviewer #1: The authors investigated predictors of subjective (self-rated, SRD) and objective difficulty (OD) of emotional facial expression perception. Based on a multi-dimensional framework, they aimed to clarify the relationship between subjective / objective difficulty of emotion recognition and (i) actor’s age and sex, (ii) observer’s age and sex, and (iii) perceived valence and arousal of the stimuli. Therefore, 658 participants (German native speakers) took part in an online survey. The final dataset included 441 participants (f : m = 312 : 129; mean age: 28.08±8.17 (females) / 29.82±8.88 (males)). Each participant rated valence and arousal of 12 four-second long videos depicting dynamic facial stimuli (total set: 480 videos; 6 female / 6 male actors; mean age of actors: 35.92; 40 different emotion categories including Ekman’s six basic emotions and 34 complex emotions). Furthermore, participants provided ratings on the Basic Emotions and Interest (BEI) scales (happiness, sadness, fear, anger, surprise, disgust, and interest) and indicated subjective difficulty of making their decisions. All ratings were given on continuous visual sliding scales. Objective difficulty was defined as the Euclidean distance from an observer’s rating to the consensus of the population on a video. This consensus was calculated as the centroid in the seven-dimensional BEI rating space. Prior to hypotheses testing, a consensus check was performed following a model for each BEI item with the rating as a dependent variable and observer age and sex as independent variables. This follows the idea of a “Ground Truth” which is not predetermined by stimulus labels, but seen as the group’s consensus. The hypotheses were examined with mixed effect models and corrected for multiple testing with the Bonferroni-Holm method. In a nutshell, this work revealed (i) more dispersed ratings and higher SRD for emotions displayed by older actors, (ii) higher OD linked to female actors, (iii) a negative quadratic effect of observer age on SRD, but no quadratic effect on OD, (iv) a higher SRD and OD in female observers, but greater target emotion attribution for women, (v) a negative quadratic relationship between valence / arousal and SRD, (vi) a negative quadratic effect of valence, but a positive quadratic effect of arousal in the OD model, and (vii) a higher predictive importance of valence for both difficulty measures compared to arousal and person-specific predictors.

The strengths of the study include:

• Novel emotion perception paradigm with dynamic facial stimuli and continuous rating dimensions rather than discrete choices

• Investigation of stimuli-specific and person-specific variables of subjective and objective difficulty of emotion perception

• Insightful discussion of the “Ground Truth” concept

The presented data appear properly organized and the statistical analysis is consistent and of high methodological quality. This work includes a careful consideration of challenges and limitations of previous literature on emotion recognition, clear-cut hypotheses and a nuanced discussion of the findings. Therefore, the manuscript deserves to be published.

Reviewer #2: The manuscript reports findings of a study investigating predictors of subjective and objective difficulty in emotion perception from dynamic facial expressions. By administrating a novel paradigm of emotions rating, authors found that both observers-related (age, sex) and stimuli-related (age, sex, arousal, valence) variables have a predictive role on subjective/objective difficulty.

The purpose of the manuscript is relevant and innovative.

However, some important methodological issues prevent the study from the necessary rigor to be acceptable for publication.

I list point-by-point my concerns:

- At lines 282-283, the authors mentioned that “The minimum sample size was set to 400 participants, which would ensure that on average every one of the 480 video clips used in the study (see Materials) was rated by 10 participants …”, that means that each participant was administered a specific sub-set of stimuli, different from other subjects (about 11 videos randomly selected). This point is especially critical considering that the design of the paradigm should be cleaned by potential biases (e.g. different videos could differently influence the answer of the participant). I suggest selecting a sub-set of videos administered to all the participants (or a sub-set of participants).

- At lines 308-309 authors reported: “participants provided ratings on the Basic Emotions and Interest (BEI) scales: happiness, sadness, fear, anger, surprise, disgust, interest”. Participants rated each video on 7 emotions, 6 basic emotions, and 1 complex emotion. I suggest authors justifying the reason why BEI includes 6 basic and 1 complex emotion in the method.

- The sample is not balanced in terms of gender distribution. I suggest including the observer’s gender as a covariate in the analyses (for hypotheses 1 and 3).

- Since the observer’s age was investigated as a predictor of subjective and objective difficulty, I believe that participants should be described also in terms of % of subjects within different ranges of age (e.g. % participants 20-30 years old, % participants 30-40 y old….)

- Authors should report the exact p-value in tables or in text, also for the explorative analyses.

- The introduction is very dispersive and difficult to read, I suggest shortening it and to focus on the background supporting the three main hypotheses of the study.

- The manuscript should be reviewed by a native-English speaker.

6. PLOS authors have the option to publish the peer review history of their article (what does this mean?). If published, this will include your full peer review and any attached files.

Reviewer #1: No

Reviewer #2: No

---

## [Author Response · Author response to Decision Letter 0]

25 Oct 2021

We thank the Reviewers for all their suggestions and comments. We included detailed responses to all raised points in the file "Response to Reviewers".

---

## [Decision Letter · Decision Letter 1]

21 Dec 2021

PONE-D-21-12940R1Subjective and Objective Difficulty of Emotional Facial Expression Perception from Dynamic StimuliPLOS ONE

Dear Dr. Matyjek,

Thank you for submitting your manuscript to PLOS ONE. After careful consideration, we feel that it has merit but does not fully meet PLOS ONE’s publication criteria as it currently stands. Therefore, we invite you to submit a revised version of the manuscript that addresses the points raised during the review process. Reviewer 1 is satisfied with your revision, but Reviewer 2 recommend rejection. I decided to give you a chance to further improve your work taking into account concerns of Reviewer 2. Please address all comments point-by-point in your rebuttal letter addressed to me.

If applicable, we recommend that you deposit your laboratory protocols in protocols.io to enhance the reproducibility of your results. Protocols.io assigns your protocol its own identifier (DOI) so that it can be cited independently in the future. For instructions see: https://journals.plos.org/plosone/s/submission-guidelines#loc-laboratory-protocols. Additionally, PLOS ONE offers an option for publishing peer-reviewed Lab Protocol articles, which describe protocols hosted on protocols.io. Read more information on sharing protocols at https://plos.org/protocols?utm_medium=editorial-emailutm_source=authorlettersutm_campaign=protocols.

We look forward to receiving your revised manuscript.

Kind regards,

Marina A. Pavlova, PhD

Academic Editor

PLOS ONE

Reviewers' comments:

Reviewer's Responses to Questions

**Comments to the Author**

1. If the authors have adequately addressed your comments raised in a previous round of review and you feel that this manuscript is now acceptable for publication, you may indicate that here to bypass the “Comments to the Author” section, enter your conflict of interest statement in the “Confidential to Editor” section, and submit your "Accept" recommendation.

Reviewer #1: All comments have been addressed

Reviewer #2: (No Response)

2. Is the manuscript technically sound, and do the data support the conclusions?

Reviewer #1: Yes

Reviewer #2: No

3. Has the statistical analysis been performed appropriately and rigorously? 

Reviewer #1: Yes

Reviewer #2: Yes

4. Have the authors made all data underlying the findings in their manuscript fully available?

Reviewer #1: Yes

Reviewer #2: Yes

5. Is the manuscript presented in an intelligible fashion and written in standard English?

Reviewer #1: Yes

Reviewer #2: Yes

6. Review Comments to the Author

Reviewer #1: The authors investigated predictors of subjective (self-rated, SRD) and objective difficulty (OD) of emotional facial expression perception. Based on a multi-dimensional framework, they aimed to clarify the relationship between subjective / objective difficulty of emotion recognition and (i) actor’s age and sex, (ii) observer’s age and sex, and (iii) perceived valence and arousal of the stimuli. In a nutshell, this work revealed (i) more dispersed ratings and higher SRD for emotions displayed by older actors, (ii) higher OD linked to female actors, (iii) a negative quadratic effect of observer age on SRD, but no quadratic effect on OD, (iv) a higher SRD and OD in female observers, but greater target emotion attribution for women, (v) a negative quadratic relationship between valence / arousal and SRD, (vi) a negative quadratic effect of valence, but a positive quadratic effect of arousal in the OD model, and (vii) a higher predictive importance of valence for both difficulty measures compared to arousal and person-specific predictors.

The strengths of the study include:

• Novel emotion perception paradigm with dynamic facial stimuli and continuous rating dimensions rather than discrete choices

• Investigation of stimuli-specific and person-specific variables of subjective and objective difficulty of emotion perception

• Insightful discussion of the “Ground Truth” concept

The presented data appear properly organized and the statistical analysis is consistent and of high methodological quality. This work includes a careful consideration of challenges and limitations of previous literature on emotion recognition, clear-cut hypotheses and a nuanced discussion of the findings. Therefore, the manuscript deserves to be published.

Reviewer #2: I thank the authors for the work on the manuscript to address our concerns.

However, my main concern, the fact that each participant judged only a small sub-set of videos, still remains critical and prevents me from estimating the manuscript suitable for publication. Although authors mentioned that whether they select only videos administered to all the sample (about 12?) the stimuli would lose variance, I believe that the results were influenced by bias related to inter-subjects differences and therefore not valid. Also, the inclusion of random intercepts for the observers and/or videos does not solve this effect. In particular, the results on objective and subjective difficulty are based on the response of about 10 subjects per video (in some cases also about 5 participants).

This concern should be solved only whether authors focus on a sub-set of videos and increase the sample for administering this sub-group of stimuli. In fact, the ideas and hypotheses of the work are relevant and innovative and need a rigorous design to verify them.

7. PLOS authors have the option to publish the peer review history of their article (what does this mean?). If published, this will include your full peer review and any attached files.

Reviewer #1: No

Reviewer #2: No

---

## [Author Response · Author response to Decision Letter 1]

26 Mar 2022

Dear Editor,

Thank you for considering our manuscript. We have included a detailed response to Reviewers in an uploaded file "Response to Reviewers.pdf".

Kind regards,

Magdalena Matyjek

---

## [Decision Letter · Decision Letter 2]

27 Apr 2022

PONE-D-21-12940R2Subjective and Objective Difficulty of Emotional Facial Expression Perception from Dynamic StimuliPLOS ONE

Dear Dr. Matyjek: Thank you for submitting your manuscript to PLOS ONE. After careful consideration, we feel that it has merit but does not fully meet PLOS ONE’s publication criteria as it currently stands. Therefore, we invite you to submit a revised version of the manuscript that addresses the points raised during the review process. As your manuscript had been initially evaluated by two Reviewers with comnpletely different outcome, I asked Reviewer 3 to have a look at your revised version. This Reviewer report that the MS is suitable for publication, but recommend that you will explicitly mention the problem/issue that elicited reservation of initial Reviewer as one of limitations of the study. Please consider doing so in your revised version.

We look forward to receiving your revised manuscript.

Kind regards,

Marina A. Pavlova, PhD

Academic Editor

PLOS ONE

Journal Requirements:

Reviewers' comments:

Reviewer's Responses to Questions

**Comments to the Author**

1. If the authors have adequately addressed your comments raised in a previous round of review and you feel that this manuscript is now acceptable for publication, you may indicate that here to bypass the “Comments to the Author” section, enter your conflict of interest statement in the “Confidential to Editor” section, and submit your "Accept" recommendation.

Reviewer #2: (No Response)

Reviewer #3: All comments have been addressed

2. Is the manuscript technically sound, and do the data support the conclusions?

Reviewer #2: No

Reviewer #3: Yes

3. Has the statistical analysis been performed appropriately and rigorously? 

Reviewer #2: N/A

Reviewer #3: I Don't Know

4. Have the authors made all data underlying the findings in their manuscript fully available?

Reviewer #2: Yes

Reviewer #3: Yes

5. Is the manuscript presented in an intelligible fashion and written in standard English?

Reviewer #2: Yes

Reviewer #3: Yes

6. Review Comments to the Author

Reviewer #2: I appreciate the effort of the authors in explaining the rationale behind the experimental design they adopted. However, my concerns still remain. Especially, I understand the purpose to maximize the external validity. Nevertheless, this should not be at the expense of the sufficient level of internal validity of a study.

Reviewer #3: Please note that I did not review the previous versions of the manuscript.

Regarding the responses of authors, It is difficult to completely exclude the problem of inter-subject bias but for me the authors have done everything possible to assess this bias and to respond to the reviewer. Maybe it would be interesting to add some elements about this possible bias in the limitations but I think that it is not an argument to reject the paper.

Regarding, the last version of the manuscript (R2), I find it very clear and interesting. I am not comfortable with the statistical analyzes used but the method seems in line with the objectives and the conclusions bring new important elements concerning the recognition of emotion in humans.

I suggest to the authors to specify what is represented in square brackets in the tables.

7. PLOS authors have the option to publish the peer review history of their article (what does this mean?). If published, this will include your full peer review and any attached files.

Reviewer #2: No

Reviewer #3: No

---

## [Author Response · Author response to Decision Letter 2]

5 May 2022

We thank the Editor for allowing us to revise our manuscript and the Reviewers for their evaluations. In the separately attached file "Response to Reviewers", we reply to the latest reviews.

---

## [Editor Report · Decision Letter 3]

17 May 2022

Subjective and Objective Difficulty of Emotional Facial Expression Perception from Dynamic Stimuli

PONE-D-21-12940R3

Dear Dr. Matyjek,

We’re pleased to inform you that your manuscript has been judged scientifically suitable for publication and will be formally accepted for publication once it meets all outstanding technical requirements.

Kind regards,

Marina A. Pavlova, PhD

Academic Editor

PLOS ONE
---

## [Editor Report · Acceptance letter]

20 May 2022

PONE-D-21-12940R3 

Subjective and Objective Difficulty of Emotional Facial Expression Perception from Dynamic Stimuli 

Dear Dr. Matyjek:

I'm pleased to inform you that your manuscript has been deemed suitable for publication in PLOS ONE. Congratulations! Your manuscript is now with our production department. 

Kind regards, 

on behalf of

Prof. Marina A. Pavlova 

Academic Editor

PLOS ONE